# Synthesis of LDHs Based on Fly-Ash and Its Influence on the Flame Retardant Properties of EVA/LDHs Composites

**DOI:** 10.3390/polym14132549

**Published:** 2022-06-23

**Authors:** Shaoquan Li, Xiao-dong Zhu, Long Li, Yi Qian, Qingjie Guo, Jingjing Ma

**Affiliations:** 1School of Chemical Engineering, Qingdao University of Science and Technology, Qingdao 266042, China; chemlsq@163.com (S.L.); xiao-dong_zhu@qust.edu.cn (X.-d.Z.); qj_guo@yahoo.com (Q.G.); 2School of Materials Science and Engineering, Guangdong University of Petrochemical Technology, Maoming 525000, China; 3School of Environment and Safety Engineering, Qingdao University of Science and Technology, Qingdao 266042, China; lilongyln@yeah.net; 4State Key Laboratory of High-Efficiency Utilization of Coal and Green Chemical Engineering, Ningxia University, Yinchuan 750021, China; majingjing@nxu.edu.cn

**Keywords:** fly-ash, ethylene-vinyl acetate, layered double hydroxides, flame retardant, smoke suppression, thermal stability

## Abstract

Fly-ash, a kind of large solid waste in energy industry, has brought about serious environmental problems and safety consequences. No efficient way has been found yet to deal with it worldwide. The focus of contemporary research are mainly placed on the reuse of aluminum and iron, but with a low utilization rate less than 30%. Having destroyed the ecological balance, fly-ash has become a challenge drawing the attention of people in the solid waste industry. In this paper, a smoke-suppressant and flame-retardant layered double hydroxide (LDH) featuring Mg-Al-Fe ternary was successfully synthesized by fly-ash after coprecipitation. XRD results presented LDHs successful synthesis. Then, exploration on the flame retarding properties of LDHs in composites composed by ethylene vinyl acetate (hereinafter referred to as EVA)/LDHs was carried out by UL-94, limiting oxygen index (LOI), cone calorimeter (CCT), smoke density (SDT), and thermogravimetry-Fourier transform infrared spectrometry (TG-IR) tests. UL testing results showed that most of the samples had a vertical combustion rating of V-0. LOI results showed the highest LOI value of ELDH-1, amounting to as high as 28.5 ± 0.1 while CCT results showed that the rate of heat releasing, mass loss, and smoke production of composite materials were decreased significantly compared with corresponding data of pure EVA. The ELDH-1 sample displayed the lowest peaks of heat release rate (pHRR) value of 178.4 ± 12.8 Kw·m^−2^ and the lowest total heat release (THR) value of 114.5 ± 0.35 KJ·m^−^^2^. Then, SDT indicated that under respective ignition and non-ignition conditions, all composite materials present a good smoke suppression performance. Additionally, digital photographs after CCT demonstrated that EVA/LDHs composites could enhance the formation of compact charred layers, and prevent their splitting, which effectively prevent the underlying materials from burning. Finally, TG-IR findings showed that compared with pure EVA, EVA/LDHs composites also achieved a higher-level thermal stability.

## 1. Introduction

Aerosol in particulate matter (PM), especially ash generated via coal combustion, greatly influences the environment, especially when linked to waste materials to be disposed [1,2]. Inhalation of fly-ash generated via coal combustion is known as one of the possible causes for lung inflammations [3]. Fly-ash, generated via powdered coal combustion in the thermal power plants, remains one of the various substances that cause pollution of air, water, and soil, disrupt the ecological cycle and give off environmental hazards [4,5,6]. A wide variety of factors, such as the composition of parent coals, the combustion situations, the efficiency, and types of emission-control devices, as well as the disposing approach adopted, determines the physical, chemical, as well as mineralogical features [7]. Fly-ash contains lots of naturally occurring elements, among which Al, Fe, Zn, and Cu are the elements generally enriched in ashes.

Layered double hydroxides (LDHs), as an inorganic compounds, are an essential class that possess a 2D sheet-style structure with bindings in each layer. LDHs, a kind of two-nanostructured material made up of metal oxide/hydroxide layers charged positively, as well as inter-layer exchangeable anions, have been proven to be provided with excellent flame retardation performance and smoke suppression properties by virtue of their incredibly special chemical composition and layered structure [8] They are also a class of anionic clay with a general formula of [M^II^_1__−__x_M^III^_x_(OH)_2_]^x+^[(Y^m−^)_x/m_] nH_2_O, with [M^II^_1−x_M^III^_x_(OH)_2_]^x+^ referring to layer cation and [(Y^m−^)_x/m_] known as compositions of interlayer anion, respectively. M^II^ refers to divalent metal cation; M^III^, a trivalent metal one; and Y, m valence inorganic or organic acid ones [9]. The generally adopted approach for the preparation of LDHs turns to be the co-precipitation of metal salts in alkaline medium in various compositions at a constant pH, which is usually followed by the hydrothermal precipitate ageing [10]. Considerable heat will be absorbed while the temperature of the burning system will be dramatically reduced during thermal decomposition. The inert carbon dioxide released and water vapor generated dilute the content of combustible gases, causing gas phase flame retardation. Moreover, given their special layer structure, the existence of the spacious surface areas and surface adsorption activated centers makes it possible for LDHs to adsorb volatile substances generated during such a process. The pyrolysis residue generated at last, including magnesium and alumina, cover the surface of the polymers and make up an insulation layer that prevents them from the heat and oxygen in the air. LDHs, as a efficient flame retardant, are endowed with diversified advantages such as their non-toxicity, non-halogenation, and no corrosive gas generation for polymers.

In recent years, some methods of preparing zeolite and LDHs absorbents with fly-ash have been reported [11,12]. Fly-ash can be used as an important raw material to synthesize LDHs by coprecipitation method instead of pure reagent. This is considered to be an important means to solve the environmental pollution of fly-ash solid waste. In Vikranth Volli’s study [13], Mg-Al LDHs with bifunctional groups was synthesized by coprecipitation method with fly-ash. In addition, the modified fly-ash can be directly used to synthesize zeolite and hydrotalcite [14]. In the research of Ruan and his collaborators, it was found that LDHs could also synthesized using activated fly-ash and ground granulated blast-furnace slag [15]. Based on the above-discussed literature, the synthesis of LDHs based on fly-ash has been gradually concerned by researchers. As we all know, as one of the most common environmentally friendly flame retardants, LDHs has been used in polymer flame retardants [16]. Combining the above two advantages, we can not only recycle solid waste of fly-ash, but also convert fly-ash into LDHs for halogen-free flame retardant, so as to achieve the purpose of waste resource utilization and improving fire safety.

Given its useful features, such as its strong adhesion to various glass substrates, low moisture absorbing, and low resin cost [17], ethylene-vinyl acetate copolymer (EVA), widely known as a transparent elastomeric material, has been widely applied to industries related to wire and cable, film, hot melt adhesive, as well as coating [18], but is not suitable to be extensively applied, since the chemical composition makes raw EVA relatively flammable. Filling additives into EVA is one general way adopted to improve its flame retardation. As a typical halogen-free flame retardant, LDHs plays an important role in the flame retardant process of EVA.

In the paper, a series of Mg-Al-Fe LDHs (Mg:(Al + Fe) = 1:1~5:1) were synthesized based on fly-ash coprecipitation, characterized by XRD, which were then applied to EVA for flame retarding. Additionally, the flame retarding features and thermal characteristics of these EVA/LDHs composites in question have been deeply explored by LOI, CCT, SDT, and TG-IR, offering a new insight that LDHs could be synthesized by fly-ash and adopted as a valid flame retarding choice for the EVA.

## 2. Experimental

### 2.1. Materials

Materials including EVA18 (containing 18 wt% vinyl acetate) bought from Beijing Eastern Petrochemical Co., Ltd. (Beijing, China) and fly-ash supplied by Changle Shengshi Thermal Power Co., Ltd. (Changle, China), containing Al (554.9 μg/g), Fe (189.8 μg/g), Zn (18.3 μg/g), Cu (0.238 μg/g), Hg (0.631 μg/g), Pb (0.399 μg/g), Ni (0.418 μg/g), Cr (0.605 μg/g), Cd (0.132 μg/g) were adopted, with the result further checked by inductively coupled plasma atomic emission spectrometry (ICP-AES). Other reagents applied were all standard laboratory reagents, and applied as what they were received with no additional purification.

### 2.2. Synthesis of LDHs

**Synthesis of LDHs by roasting reduction method.** Fly-ash that was dried at 100 °C for 4 h, was ground by a Ball Machine and split into 200-mesh-pass particles. Then, fly-ash, along with MgO, was blended with Mg/(Al + Fe) = 3.0/1.0, after which, the mixture was roasted in a muffle furnace at 550 °C for 6 h, the acquired mixture was then leached in the Na_2_CO_3_ solution acquired via [CO_3_^2−^]/([Al^3+^ + Fe^3+^]) = 2.0. Each blending process was added to the emulsifying machine whose rotor speed was 200 r min^−1^ for 10 h simultaneously. The sample prepared was therefore named FA-MgO.

**Synthesis of LDHs by coprecipitation method.** Fly-ash was prepared after filtration, which, together with MgCl_2_·6H_2_O, was then mixed with M^2+^/M^3+^ molar ratios of 1:1~5:1 (Mg:(Al + Fe) = 1:1~5:1) (Solution A). The strong base solution containing 0.4 mol·L^−1^ Na_2_CO_3_, and 1.5 mol·L^−1^ NaOH. (Solution B) was prepared and added at the same speed to a three-necked and round-bottomed flask equipped with a mechanical stirrer, as well as a water bath, together with solution A. Corresponding stirring speed remained the same at 80 °C. After the reaction was completed, the slurry acquired was then filtered, thoroughly washed, and dried at a temperature of 80 °C to get Mg-Al-Fe-LDHs. The sample prepared was named LDH1-LDH5.

### 2.3. Preparation of the EVA Composites

Composites were melt-compounded with a mixer at about 120 °C for 10 min, after which, the mixture was compression-molded at the temperature about 120 °C into sheets which were initially cut into properly sized specimens for the following burning test under 10 MPa for 10 min. The additive level of all samples in this paper was 50%, with the samples named as EVA (pure EVA), EFMgO (50% FA-MgO), ELDH-0 (50% pure fly-ash), ELDH-1 (50% LDH1), ELDH-2 (50% LDH2), ELDH-3 (50% LDH3), ELDH-4 (50% LDH4), and ELDH-5 (50% LDH5), respectively.

### 2.4. Characterization

**X-ray diffraction (XRD).** XRD results were at room temperature on a Philips X’Pert Panalytical diffractometer following Cu-Kα radiation (λ = 0.1542 nm).

**Scanning electron microscopy (SEM).** The SEM images acquired using a JSM-6700F instrument (Japan) with a parameter condition of 5 kV.

**Cone calorimeter test (CCT).** The EVA composites were determined with cone calorimeter test (CCT) by the JCZ-2 cone calorimeter (China) in accordance with ISO 5660 standard. The EVA composites whose size was 100 × 100 × 3 mm^3^ were wrapped in aluminum foil, placed horizontally on the sample rack, and heated with an external heat source of 50 kW·m^−2^.

**Raman spectra.** The char residual of EVA composites were evaluated by Invia Qontor Raman spectrometer (Renishaw Co., Ltd., England).

**Limiting oxygen index (LOI).** LOI was subjected to ASTM D 2863, an HC-2 oxygen index meter (Jiangning Analysis Instrument Company, China). The dimensions of specimens applied were 100 × 6.5 × 3 mm^3^.

**UL-94 testing.** UL-94 was subjected to a model CFZ-II horizontal and vertical burning tester (Jiangning Analysis Instrument Company, China) according to the ASTM D 3801 (GB/T 2408).

**Smoke density test (SDT).** JQMY-2 (Jianqiao Co., China), a smoke density test machine, was applied to measure the smoke characteristics in accordance with ISO 5659-2 (2006). The dimensions 75 × 75 × 2.5 mm^3^ of each specimen was wrapped in aluminum foil and exposed horizontally to an external heat flux of 25 kW·m^−2^ with or without the application of a pilot flame.

**Thermogravimetry-Fourier transform infrared spectrometry (TG-IR).** Thermogravimetric-infrared analysis (TG-FTIR) was performed on a DT-50 instrument at a heating rate of 20 °C·min^−1^ (N_2_ atmosphere), and the temperature range was 40–800 °C.

**Tensile test.** Tensile test was completed according to the procedure in GB/T 1040.1-2006 using a testing machine.

## 3. Results and Discussion

### 3.1. XRD Characterization of Materials

XRD patterns of the materials are illustrated in Figure 1 while the lattice parameters are shown in Table 1. Diffraction peaks of (003), (006), (009), (015), (018), (110), and (113) of all samples are in perfect line with layered structures, and result from LDHs indexed by the JCPDS X-ray powder diffraction file of No. 38-0487, proving the successful synthesis of LDHs based on fly-ash [19,20,21,22], but there were not the same peaks in the fly-ash XRD pattern. In addition, fly-ash with complex composition fails to synthesize highly purified LDHs, so the corresponding diffraction peaks are also shown in Figure 1. Generally, as shown in Figure 1, the diffraction peaks of all Mg-Al-Fe LDHs samples are narrow and sharp while the baselines remain low, which further proves the well-established LDHs crystallinity; in the region with a low angle, 2θ values of (003), (006), and (009) crystal planes have good multiple relationship, which proves the outstanding layered structure of synthesized LDHs; diffraction peaks of the (110) as well as (113) crystal ones are obviously separated near 60°, indicating that high regularity degree of negative ions among layers, and the good symmetry of synthesized LDHs. Meanwhile, connected with Table 1, *d*_(003)_ of six samples are 0.770 nm, 0.773 nm, 0.789 nm, 0.784 nm, and 0.769 nm, respectively, showing that negative ions among layers of the synthesized LDHs are actually carbonate anion.

### 3.2. Performance Test of EVA/LDHs Composites

#### 3.2.1. CCT of EVA/LDHs Composites

CCT, complying with the principle of oxygen consumption, is frequently adopted to assess the fire reaction behavior of polymer materials [23]. Though performed on a small scale, the cone calorimeter test provides much information concerning the combustion performance, especially when HRR and pHRR applied to estimate the corresponding intensity as an essential parameter [24,25,26,27]. The results are found to be closely linked to those acquired from a large-scale combustion test, further highlighting the test value to estimate the combustion performance of materials in an actual fire [28,29]. In this study, the same sample was repeated five times about the cone testing, and the experimental data and charts in the manuscript were all selected in the middle group. The specific error range is showed in Table 2.

**HRR of LDHs/EVA composites.** HRR, an essential parameter to express fire intensity, and pHRR, an essential parameter to evaluate the intensity of real fire, not only reflect the gas heat generated during the combustion by polymer decomposition, but also show the released heat, contributing to the judgement over fire spreading and its threats to the people [30,31]. HRR curves for all the samples are shown in Figure 2 while crucial numerical results of the measurements are listed in Table 2. Given the two-step pyrolysis performance of the EVAs, each sample embraces two HRR peaks during the combustion.

It is shown that the HRR curves of pure EVA rose up quickly when ignited, reaching the pHRR of 1717.9 ± 21.4 Kw·m^−2^ in only 195 ± 12 s. When 50 wt% LDH1 was added to EVA (ELDH-1), the curves showed a pHRR of 178.40 ± 12.8 Kw·m^−2^, also a lowest HRR of ELDH-1 among all samples. It is found in Figure 2 that HRR of all samples declined, compared with that of pure EVA. So it could be concluded that a suitable amount of fly-ash has considerable effect on the improvement of flame retardation. A possible reason turns out to be that when formed, the char layered system performs as a protective barrier which limits oxygen diffusion to the substrate and retards the volatilization of the flammable decomposition products. In addition, the released inert carbon dioxide and water vapor will dilute the concentration of combustible gas, and this charred layer prevents heat transfer from the surface and the melting polymer. Thus, HRR of all EVA/LDHs samples is reduced.

**THR of****EVA/LDHs composites.** THR, sum of the heat release by the material from light until the flame extinguishes [32], is only related to the internal energy of the material to a certain extent, but independent from environmental factors. It can be concluded from Table 2 that ELDH-1 has the lowest THR value of 114.5 ± 0.35 KJ·m^−2^. As shown in Figure 3, pure EVA releases the most heat in the combustion process, while the heat release of ELDH-1 to ELDH-5 has a decreasing tendency, which shows that LDHs reduce the total heat release. The slope of THR curve shows that ELDH-1 possesses the slowest slope, indicating its slow burning speed attributed to LDHs migration onto the sample surfaces, which can be the barrier from the flame zone to the underlying materials, and limits the flammable gases to the flame zone. At the same time, LDHs absorb considerable heat during the thermal decomposition, with the combustion temperature reduced. In addition, the pyrolysis residue, made up of magnesium and alumina, cover the surface of the polymer in question and establish an insulating layer that insulates heat and oxygen from the air, LDHs contributing to the reduction of the total heat release.

**Mass of****EVA/LDHs composites.** Dynamic curves, mass versus time, of the above eight samples are shown in Figure 4, where the weight variation of the char residue is also presented. Pure EVA was almost completely consumed. The curve of ELDH-0~ELDH-5 mass-losing will slow down and their residual mass are all kept above 30%, which proves that LDHs improve flame retardation of composite materials. As the combustion begins, a charred layer is formed on the EVA/LDHs surfaces, which slows down the combustion process. LDHs assist the burning of composite materials so as to create char residue, isolating O_2_ and heat transfer between burning areas and the bottom of the carbon layer. Complete char residue may occur on the surface of the burning sample during the combustion. Such a physical process tends to perform like a protecting barrier. Along the combustion, the charred layer gradually breaks, causing the mass loss of the composites [33].

**Digital photos of char residue.**Figure 5 presents digital photos of char residue of EVA/LDHs composites. It can be seen that there is no visible char residua the pure EVA sample while charred residues left in the ELDH-0~ELDH-5 samples are significantly integrated as well as compact. Such a result is in line with data shown in Figure 4. LDHs, a material featuring its pore structure, is provided with excellent adsorption as well as flame retardation capacity, which increase the efficiency of flame retarding, and control the smoke generation. Furthermore, LDHs lead to the formation of a compact layer on the sample surface, which means that LDHs improve the structure of the charred layers and strengthen their thermal stability, guaranteeing better flame retardation and smoke suppression of flame retardant. The photographs after cone calorimeter test are well agreed with data shown in Figure 3 and Figure 4. The char structure helps introduce the combustion of the flame retarding EVA composites. The formation of efficient ceramic materials prevents the transfer of heat mass between the flame zone as well as the burning substrate, protect the potential materials from continuously burning and retard the polymer cracking.

**SEM of char residue.** SEM analysis was applied to examine the char residue that were after CCT test for char appearance to explore how char structure determines its fire resisting properties. Figure 6 shows the SEM photographs of the upper and lower surfaces of the charred residues of EFMgO and ELDH-1 samples magnified 5000×. Its effective protection obviously have the flame retardation efficiency improved during the combustion. As is further described in the picture, the EFMgO surface is fragile and cracked. In addition, char residue of ELDH-1 displays a more tight structure. Compact char structure shows that EVA/LDHs composites could prevent both heat and mass transfer between the flame zone and the materials, which is of great significance for EVA application. In addition, SEM images also explain the flame retarding performance of EVA/LDHs composites in CCT test.

**R****aman spectra of char residue.** Raman spectra is an important test method to study the flame retardant mechanism of composite materials. It can be clearly seen from Figure 7 that EFMgO and ELDH-1 have two characteristic peaks at 1358 cm^−1^ and 1598 cm^−1^, corresponding to the D band and G band, respectively. The area ratio of the D band and G band (I_D_/I_G_) generally represents the carbonization degree of char residue. When the I_D_/I_G_ value is lower, the char residue structure is denser [24,25]. It is obvious from Figure 7 that ELDH-1 has a lower I_D_/I_G_ value than EFMgO, indicating that the char residue structure of ELDH-1 is denser. The LDHs based on fly-ash can contribute to the formation of carbon layer.

#### 3.2.2. SDT of EVA/LDHs Composites

SDT offers more information about smoke generating in detail when CCT test reflects the combustion performance of the samples which plays an essential role in the evaluation of the smoke suppression properties of the samples.

Luminous Flux is used to evaluate the amount of smoke production. Luminous Flux patterns of the samples are shown in Figure 8, where curve a(without flame) and curve b (with flame) both presented a sharp pure EVA and EFMgO decline that stabilized at a low level at last. A slower Luminous Flux curve trend of ELDH-1~ELDH-5 can be seen when compared with that of pure EVA, indicating that LDHs are endowed with smoke suppression properties. Luminous Flux of ELDH-5 stabilized at 80% above when not ignited, evidently better than that of EVA. Figure 8b shows that Luminous Flux of ELDH-2~ELDH-5 has been more than 65%, especially ELDH-3, which is over 85%, showing that almost no smoke was spilled out along the combustion. The smoke suppression became increasingly apparent when LDHs were added to EVA/LDHs. LDHs are decomposed into Al_2_O_3_ and MgO when heated at a high temperature, which prevent thermal transport and air inside the materials. These processes change the thermal decomposition pathways of polymers, contributing to the cross-linking char formation. Al_2_O_3_ and MgO promote the formation of carbon layer, acting as a polymer framework with the property of smoke suppression. Additionally, its giant surface area, also the porous structure endows LDHs with useful features such as an excellent absorptive capacity to effectively absorb smoke [34,35].

#### 3.2.3. LOI and UL-94 Test of EVA/LDHs Composites

LOI is a simplified, also efficient method for the flammability evaluation of polymers, particularly for the screening of flame retarding polymers formulations [36]. As shown in Table 3, the LOI value of pure EVA remained as low as 19.8 ± 0.1%, which rose to 21.5 ± 0.1% when 50% of fly-ash was added, highlighting the flame retardation of fly-ash on EVA. That of all EVA/LDHs samples rose over 26.5%, which indicates that LDHs samples based on fly-ash enhance the flame retardant performance of EVA. Among these EVA/LDHs samples, LOI of ELDH-1 turned to be the highest, reaching 28.5 ± 0.1%. This phenomenon might be explained by the fact that LDHs place a synergistic effect on EVA. The LOI value increased when a proper amount of LDHs were added to the composites. UL-94 testing is an important method to evaluate the flame retardancy of polymers. The results are listed in Table 3. The results showed that most of the samples had a vertical combustion rating of V-0, the flame retardant properties of composite materials are outstanding. In addition, no dripping happened to composite materials in all these tests, which manifests that LDHs materials have the anti-dripping properties of flame retarding materials improved.

#### 3.2.4. TG-FTIR Characterization of the EVA/LDHs Composites

The TG-FTIR analysis is generally adopted to explore the thermal degrading performance of flame retarding materials, and analyzes changes in gases generated at different temperatures. The TG as well as DTG curves of EVA, EFMgO, ELDH-1, and ELDH-2 in a nitrogen atmosphere is shown in Figure 9.

**TG analysis.** When applied as a flame retardant, the thermal stability of polymers plays an essential role, which is closely linked to the release of decomposition products and char formation. Mass loss of polymers was caused by the volatilization of products generated by thermal decomposition, which was monitored as the function of a temperature ramp. The related results are given in Figure 9 and Table 4. EVA in Figure 9 experienced two degrading steps, the first of which was the loss of carbonates while the second was the remaining materials’ random chain scission, forming various unsaturated vapor elements, such as butene and ethylene [33,34]. Meanwhile, there were three weight-loss steps for ELDH-1 and ELDH-2. The first was the loss of the absorbed water in LDHs on the surface of the composites while the second and third steps were simultaneous LDHs dihydroxylation, as well as decarbonation, respectively, overlapped by acetate groups decomposition in side chains, as well as the scission of main EVA chains [37,38].

It should also be noted that ELDH-1, compared with EVA, presented a lower decomposition rate in the third step and a higher one in the first and second step. The incorporation of LDHs lowered the decomposition rate in the third step, but accelerated the loss of carbonates. No residue was left in the EVA sample at a temperature higher than 600 °C, but there was still 30% left both in the samples of ELDH-1 and ELDH-2. ELDH-1, as a binary composite, these synergistic charring effect LDHs is beneficial to improving the thermal stability of EVA composites. This result confirms their synergistic effects, which were also ascertained by the above cone calorimeter tests. Compared with char quantity, char morphology matters more in its flame retardant property. The same conclusion was drawn out in Weil and Pate’s study [39].

**FTIR characterization.** During the thermal degrading process, volatilized composite products emerging were characterized by the TG-IR technique as illustrated in Figure 10 which shows the 3D TG-IR spectra of pyrolysis products of the composites during that process and Figure 11 which depicts the FTIR spectra of pyrolysis products of the composites at different temperatures. It can be seen in Figure 11 that gas products generated in these four samples exhibited characteristic bands of 950–1150, 1250–1500, 1700–1860, 2250–2400, 2800–3150, and 3400–4000 cm^−1^, respectively. The spectra perfectly tallies with the reported FTIR characteristics of gas products, such as carboxylic acid (1700–1850 cm^−1^), CO (2250–2300 cm^−1^), CO_2_ (2300–2400 cm^−1^), and aliphatic hydrocarbons (950–1150, 1250–1500, 2800–3150 cm^−1^) [40,41]. The main decomposition products are carboxylic acid, CO, CO_2_, and aliphatic hydrocarbons in this study.

Different pyrolysis products of the composites during the thermal degradation highlighted the distinct difference in the degradation processes of these four samples. Obviously, pure EVA decomposed quickly when it was heated while the decomposition of EFMgO containing fly-ash produced more carboxylic acid and aliphatic hydrocarbons when compared with that of other samples, proving that fly-ash is not a proper flame retarding choice for EVA. However, ELDH-1 and ELDH-2 produced more CO_2_ when compared with other samples, which proves their better flame retardation when compared with other samples.

Further detailed information related to the FTIR spectra of pyrolysis products at various temperatures during thermal degrading is listed in Figure 11, where there were no peaks in CO_2_ release for pure EVA until the temperature rose to about 400 °C, except a CO_2_ peak at the temperature about 340 °C for both ELDH-1 and ELDH-2. The CO_2_ peak for ELDH-1 and ELDH-2 may mainly result from the CO_3_^2−^ in LDHs, which will transform into CO_2_ when heated.

Much carboxylic acid could be noticed to evolve from pure EVA and EFMgO, but not ELDH-1 and ELDH-2. Given LDHs’ reaction with carboxylic acid to compose H_2_O, ELDH-1 presented a small peak at 330 °C, making it possible for the carboxylic acid release to be kept at a low level. The generated carboxylic acid reflected EVA decarboxylation and the release of aliphatic hydrocarbons could be used to evaluate the break of the main chain. A sharp peak could be seen clearly for pure EVA, as shown in Figure 10 and Figure 11. The release of aliphatic hydrocarbons decreased significantly when LDHs were added. It can therefore be concluded to some extent that the synergistic mechanism between EVA and LDHs is not the gas phase process but the condensed phase process [42].

**Mechanical properties.** A large number of additive retardants will affect the mechanical properties of EVA to a certain extent. The mechanical properties of the EVA composites are shown in Table 5. The results show that the mechanical properties of all EVA composites are slightly lower than that of pure EVA. Moreover, we found that the tensile strength and elongation at break of the composites were both maintained above 80%, which may be due to the unique double layered of LDHs that makes the flame retardant melt and disperse in EVA. On the one hand, the strong chemical bonds are formed between LDHs lamellar structure and EVA resin molecules through melting blending [22]. On the other hand, the mechanical properties were affected by the dispersion of inorganic filler in the EVA matrix, the desired dispersion of LDHs resulted in greater interface area and this provided the essential condition for excellent interactions between LDHs and EVA [43]. So the mechanical properties have little influence. Therefore, there is no apparent damage to mechanical properties of EVA matrix resin while improving its flame retardant.

## 4. Conclusions

This paper proposed a new dimension for a range of fly-ash applications, with Mg-Al-Fe LDHs prepared via fly-ash coprecipitation, which were characterized by XRD. Mg-Al-Fe LDHs, whose structure was better than that of pure fly-ash, were successfully synthesized. LDHs synthesis based on fly-ash was proven to be a promising way for the practical application of fly-ash.

Comparison between the flammability and thermal degradation performance of EVA/LDHs composites and that of pure EVA and EFMgO was conducted via LOI, CCT, SDT, and TG-FTIR analysis. Results confirmed the effects placed by LDHs on the flame and thermal stability of the composites. With LDHs added to pure EVA, the LOI value increased significantly, and no dripping phenomenon happened. The CCT data indicated that HRR of the ELDH-1 was almost the lowest while the SDT results showed that EVA/LDHs composites could greatly enhance the smoke suppression. The TG-FTIR results proved the improvement of the thermal stability of EVA/LDHs while the synergistic mechanism between fly-ash and LDHs was mainly determined by the condensed phase process.

To conclude, this work offers an outstanding flame retardation performance for the application of fly-ash. This discovery is believed to improve contemporary understanding of fly-ash in the field of flame retardant.

## Figures and Tables

**Figure 1 polymers-14-02549-f001:**
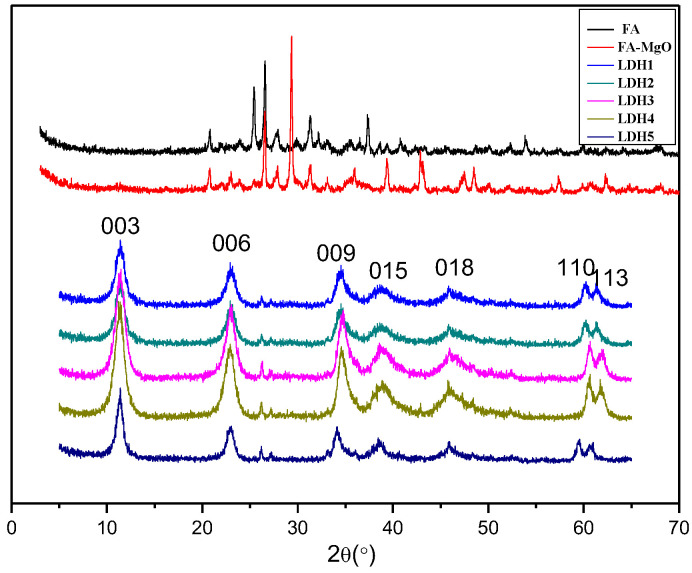
XRD pattern of samples.

**Figure 2 polymers-14-02549-f002:**
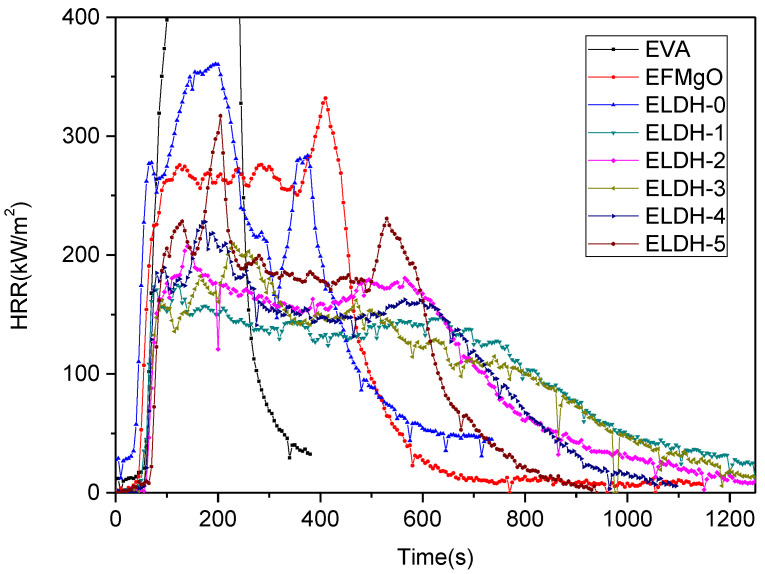
HRR curves of all EVA/LDHs composites.

**Figure 3 polymers-14-02549-f003:**
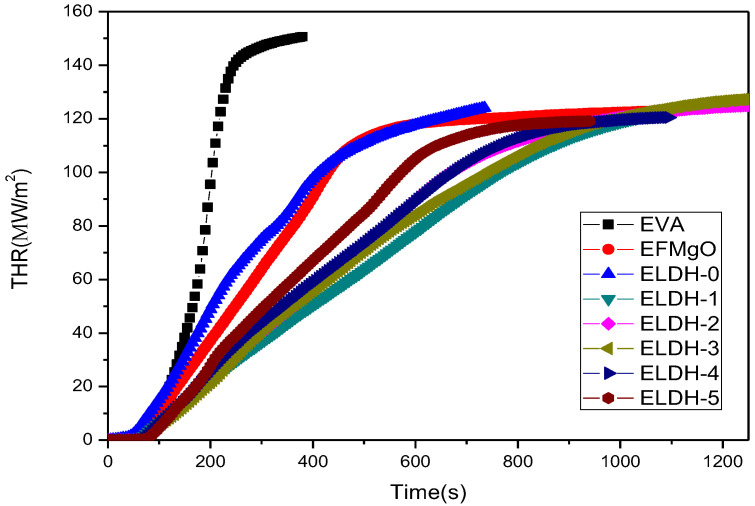
THR of the EVA/LDHs composites.

**Figure 4 polymers-14-02549-f004:**
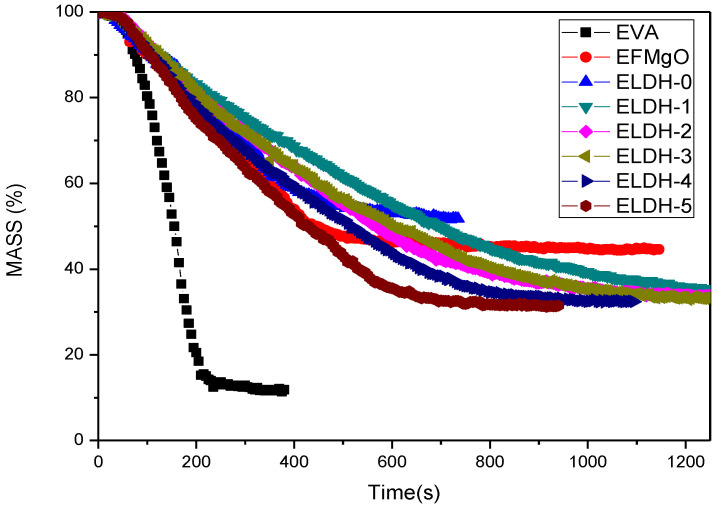
Mass of the EVA/LDHs composites.

**Figure 5 polymers-14-02549-f005:**
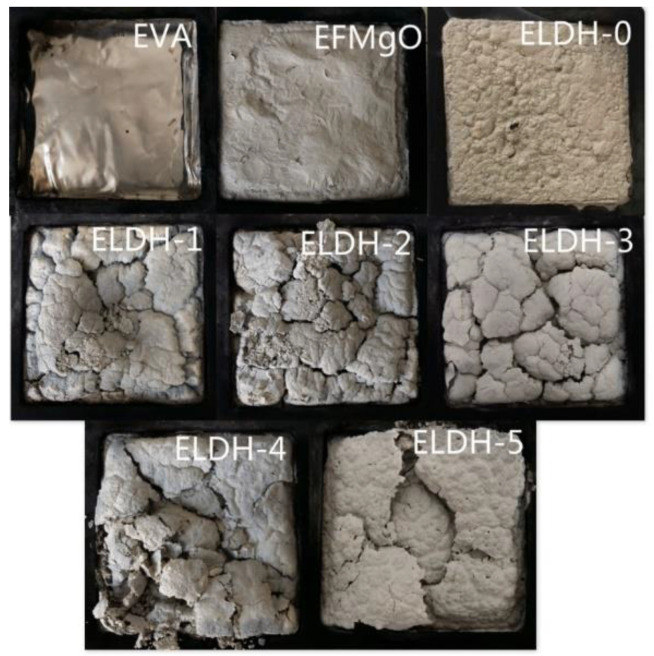
Photographs after cone calorimeter test.

**Figure 6 polymers-14-02549-f006:**
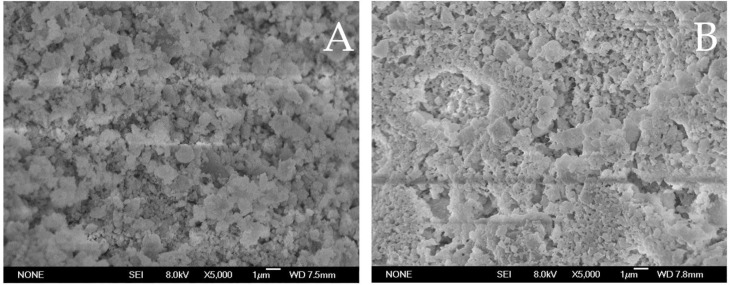
SEM images of the outer surface of the char residue ((**A**): EFMgO; (**B**): ELDH-1).

**Figure 7 polymers-14-02549-f007:**
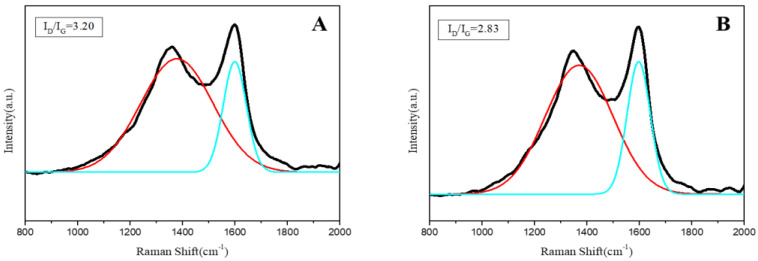
Raman spectra of the char residue ((**A**): EFMgO; (**B**): ELDH1).

**Figure 8 polymers-14-02549-f008:**
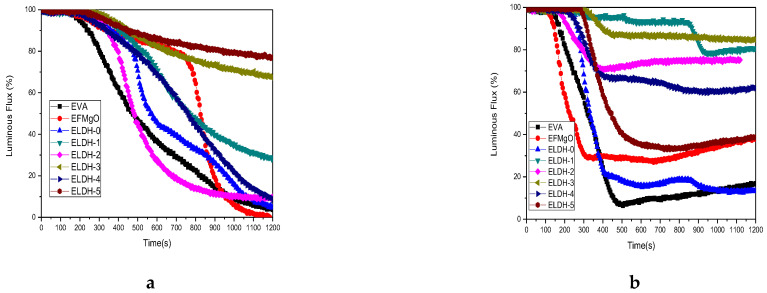
Luminous Flux curves of all EVA/LDHs Composites: (**a**) without and (**b**) with the application of the pilot flame.

**Figure 9 polymers-14-02549-f009:**
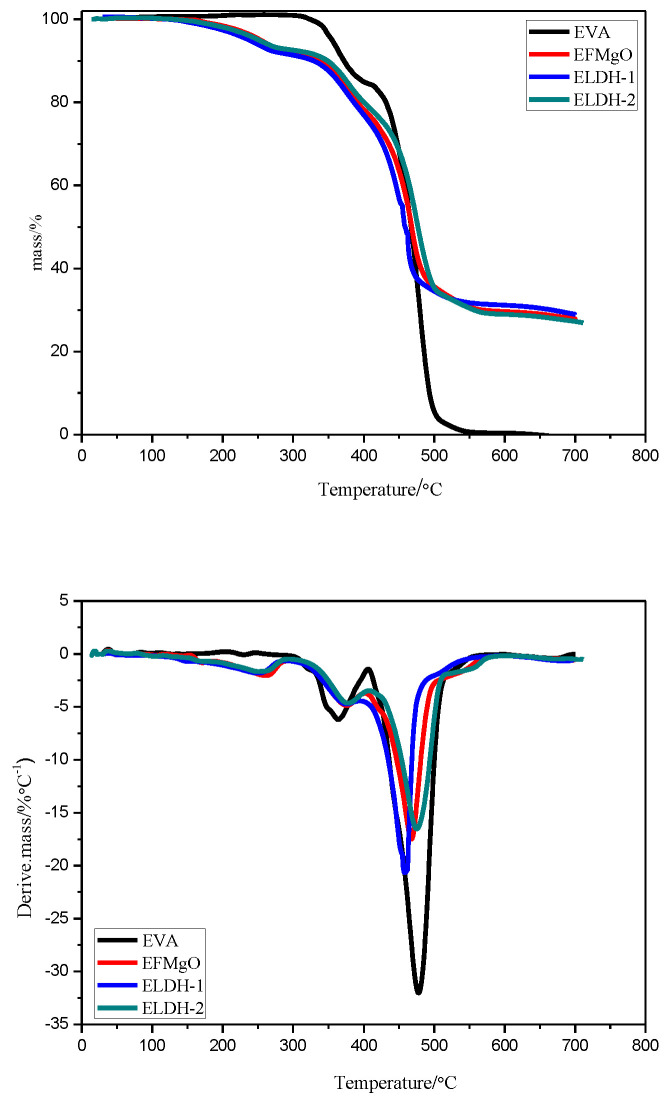
TG and DTG curves of the EVA, EFMgO, ELDH-1, and ELDH-2 composites.

**Figure 10 polymers-14-02549-f010:**
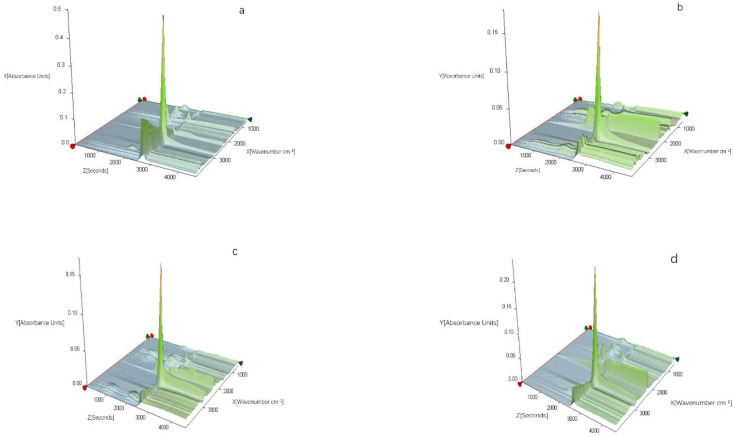
Three-dimensional TG-FTIR spectra of pyrolysis products of: (**a**) EVA, (**b**) EFMgO, (**c**,**d**) ELDH-1 and ELDH-2 samples during thermal degradation.

**Figure 11 polymers-14-02549-f011:**
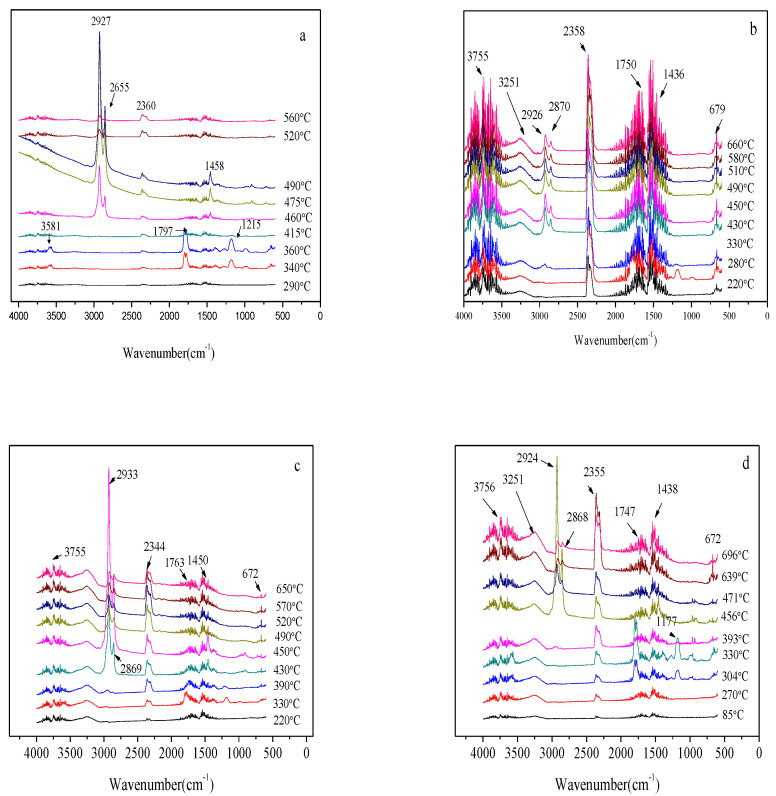
FTIR spectra of pyrolysis products of: (**a**) EVA, (**b**) EFMgO, (**c**,**d**) ELDH-1 and ELDH-2 samples at different temperatures.

**Table 1 polymers-14-02549-t001:** Structural parameters of Mg-Al-Fe LDHs.

Sample Code	LDH1	LDH2	LDH3	LDH4	LDH5
*d*_(003)_, nm	0.770	0.773	0.789	0.784	0.769
*d*_(006)_, nm	0.380	0.387	0.389	0.386	0.370
*d*_(009)_, nm	0.244	0.246	0.247	0.246	0.241
*d*_(015)_, nm	0.234	0.234	0.235	0.234	0.230
*d*_(110)_, nm	0.150	0.150	0.152	0.152	0.149
a, nm	0.300	0.300	0.304	0.304	0.298
c, nm	2.310	2.319	2.367	2.352	2.307

**Table 2 polymers-14-02549-t002:** Data from cone calorimeter test.

Sample Code	pHRR (Kw·m^−2^)	THR (MJ·m^−2^)	Time to pHRR (s)	Time to Flame Out (s)
EVA	1717.9 ± 21.4	150.6 ± 0.52	195 ± 12	302 ± 12
EFMgO	331.9 ± 17.9	123.5 ± 0.46	410 ± 9	575 ± 15
ELDH-0	360.6 ± 19.6	124.1 ± 0.33	195 ± 7	581 ± 21
ELDH-1	178.4 ± 12.8	114.5 ± 0.35	120 ± 10	1218 ± 13
ELDH-2	207.2 ± 18.9	126.5 ± 0.41	145 ± 14	1033 ± 18
ELDH-3	208.9 ± 18.3	128.2 ± 0.28	230 ± 8	1142 ± 20
ELDH-4	227.8 ± 20.6	120.4 ± 0.50	175 ± 12	900 ± 26
ELDH-5	317.1 ± 23.5	119.1 ± 0.24	205 ± 12	806 ± 28

**Table 3 polymers-14-02549-t003:** LOI values and UL-94 results for EVA/LDHs composites.

Sample Code	LOI(%)	Rating	Dripping Behavior
EVA	19.8 ± 0.1	---	No dripping
EFMgO	21.5 ± 0.1	V-2	Dripping
ELDH-0	21.6 ± 0	V-0	Dripping
ELDH-1	28.5 ± 0.1	V-0	Dripping
ELDH-2	27.5 ± 0.1	V-0	Dripping
ELDH-3	27.2 ± 0.1	V-0	Dripping
ELDH-4	26.8 ± 0	V-0	Dripping
ELDH-5	26.5 ± 0.1	V-1	Dripping

**Table 4 polymers-14-02549-t004:** The TG results of the EVA, EFMgO, ELDH-1, and ELDH-2 composites under nitrogen condition.

Sample Code	T_-5_ (°C)	T_-50_ (°C)	T_-max_ (°C)		
			Step 1	Step 2	Step 3
EVA	361.9 ± 12.4	473.7 ± 10.2	356.7 ± 5.9	481.3 ± 9.2	---
EFMgO	255.2 ± 12.9	470.2 ± 9.6	261.8 ± 4.6	373.5 ± 7.6	470.2 ± 11.5
ELDH-1	247.3 ± 14.8	452.6 ± 8.8	252.3 ± 5.2	373.1 ± 7.5	452.6 ± 10.6
ELDH-2	251.6 ± 13.5	480.3 ± 12.4	252.8 ± 6.1	375.2 ± 8.3	480.3 ± 13.2

**Table 5 polymers-14-02549-t005:** Mechanical properties for EVA/LDHs composites.

Sample Code	Tensile Strength (MPa)	Elongation at Break (%)
EVA	11.8 ± 0.4	363.8 ± 22.4
EFMgO	9.2 ± 0.3	300.6 ± 19.2
ELDH-0	9.7 ± 0.2	314.6 ± 19.6
ELDH-1	10.1 ± 0.3	327.7 ± 18.6
ELDH-2	9.9 ± 0.3	324.5 ± 18.0
ELDH-3	9.4 ± 0.4	318.4 ± 20.3
ELDH-4	9.3 ± 0.2	298.6 ± 19.8
ELDH-5	9.2 ± 0.2	298.2 ± 16.5

## Data Availability

The data presented in this study are available on request from the corresponding author.

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
