# Peer review of "Synthesis of LDHs Based on Fly-Ash and Its Influence on the Flame Retardant Properties of EVA/LDHs Composites"

_polymers, 2022, doi:10.3390/polym14132549_

Round 1

Reviewer 1 Report

I don't have any additional comments after the revision.

Author Response

The reviewe don't have any additional comments after the revision.

Reviewer 2 Report

This manuscript can be accepted.

Author Response

The reviewe don't have any additional comments after the revision.

This manuscript is a resubmission of an earlier submission. The following is a list of the peer review reports and author responses from that submission.

Round 1

Reviewer 1 Report

The comments are provided in the attachment.

Reviewer 2 Report

The idea of the article is excellent. However there is quite a lot of chemical  inconcruities, which must be corrected. Namely:

p. 2 - "...inorganic or organic acid ones" - the particles in question are not cations, they are ANIONS and can not be listed as congeneric in this row

PH - should be changed into pH

p. 4 "carbonate radical" - carbonate is not a radical in this structure, it is anion

p.5 "alumina oxide" - alumina itself means aluminium oxide

p.9 "loss of carboxylic acid" - loss of carbonates, obviously?

I would recommend to give the whole manuscript to a chemist for thorough reading

The syntheses in EXPERIMENTAL are described insufficiently, especially the second one. It should be rewritten. What did serve as a source of magnesium? What was the composition of the reactants mixture? All the reactants should be listed.

The quality of the Figure 2 is low, the background curve for the ash is indistinguishable. The authors describe "the slowest slope". However the curves vary their directions and slopes.  At what points of the curves the discussion is focused?

In the discussion of THR date the reference to a figure is absent (Fig. 3, presumably?)

The following sentence in p. 9 is disorienting: "...which indicates that LDHs samples enhance the composite flammability". Meanwhile the publication is devoted the flame retardation properties of LDH.

In TG discussion  (p.9) the statement "there was still 30% left both in the samples of ELDH-1 and ELDH-2" is inaccurate. Of course, there will be always a remaining mineral part after the polymer is burned out. It does not mean flame retardation for the polymer, it just means that incombustionable substance was added into the composition.

The references 12, 13, 14, 15, 20 are exessive and do not relate to the content of the manuscript

Round 2

Reviewer 1 Report

The manuscript can be accepted.

Reviewer 2 Report

No comments. All corrections are inserted.